# Fine-Grained Zero-Shot Learning with DNA as Side Information

**Sarkhan Badirli**
Department of Computer Science
Purdue University, West Lafayette, IN, USA
Indiana University-Purdue University Indianapolis, IN, USA

**Zeynep Akata**
University of Tübingen
Max Planck Institute for Informatics
Max Planck Institute for Intelligent Systems

**George Mohler**
Computer and Information Science Department
Indiana University - Purdue University
Indianapolis, IN, USA

**Christine J. Picard**
Department of Biology
Indiana University - Purdue University
Indianapolis, IN, USA

**Murat Dundar**
Computer and Information Science Department
Indiana University - Purdue University
Indianapolis, IN, USA
`mdundar@iupui.edu`

## Abstract

Fine-grained zero-shot learning task requires some form of side-information to transfer discriminative information from seen to unseen classes. As manually annotated visual attributes are extremely costly and often impractical to obtain for a large number of classes, in this study we use DNA as side information for the first time for fine-grained zero-shot classification of species. Mitochondrial DNA plays an important role as a genetic marker in evolutionary biology and has been used to achieve near perfect accuracy in species classification of living organisms. We implement a simple hierarchical Bayesian model that uses DNA information to establish the hierarchy in the image space and employs local priors to define surrogate classes for unseen ones. On the benchmark CUB dataset we show that DNA can be equally promising, yet in general a more accessible alternative than word vectors as a side information. This is especially important as obtaining robust word representations for fine-grained species names is not a practicable goal when information about these species in free-form text is limited. On a newly compiled fine-grained insect dataset that uses DNA information from over a thousand species we show that the Bayesian approach outperforms state-of-the-art by a wide margin.

## 1 Introduction

Fine-grained species classification is essential in monitoring biodiversity. Diversity of life is the central tenet to biology and preserving biodiversity is key to a more sustainable life. Monitoring biodiversity requires identifying living organisms at the lowest taxonomic level possible. The traditional approach to identification uses published morphological dichotomous keys to identify the collected sample. This identification involves a tedious process of manually assessing the presence or absence of a long list of morphological traits arranged at hierarchical levels. The analysis is often performed in a laboratory setting by a well-trained human taxonomist and is difficult to do at scale. Fortunately, advances in technology have addressed this challenge to some extent through the use of

35th Conference on Neural Information Processing Systems (NeurIPS 2021).

DNA barcodes. DNA barcoding is a technique that uses a short section of DNA from a specific gene, such as *cytochrome C oxidase I (COI)*, found in mitochondrial DNA, and offers specific information about speciation in living organisms and can achieve nearly perfect classification accuracy at the species level (26; 17).

As it is costly to obtain the label information for fine-grained image classification of species, Zero-Shot Learning (ZSL) that handles missing label information is a suitable task. In ZSL, side information is used to associate seen and unseen classes. Popular choices for side-information are manually annotated attributes (21; 12), word embeddings (41; 14; 27) derived from free-form text or the WordNet hierarchy (28; 2). It is often assumed that an exhaustive list of visual attributes characterizing all object classes (both *seen* and *unseen*) can be determined based only on seen classes. However, taking insects as our object classes, if no seen class species have antennae, the attribute list may not contain *antenna*, which may in fact be necessary to distinguish unseen species. In the United States alone, more than 40% of all insect species (>70,000) remain undescribed (42), which is a clear sign of the limitations of existing identification techniques that rely on visual attributes. Similarly, free-form text is unlikely to contain sufficiently descriptive information about fine-grained objects to generate discriminative vector embeddings. For example, *tiger beetle* is a class in the ImageNet dataset. However, the *tiger beetle* group itself contains thousands of known species and the Wikipedia pages for these species either do not exist or are limited to short text that does not necessarily contain any information about species' morphological characteristics. WordNet hierarchy may not be useful either as most of the species names do not exist in WordNet.

Given that DNA information can be readily available for training (35; 36), species-level DNA information can be used as highly specific side information to replace high-level semantic information in ZSL. For seen classes, species-level DNA information can be obtained by finding the consensus nucleotide sequence among samples of a given species or by averaging corresponding sequence embeddings of samples. For unseen classes, species-level DNA information can be obtained from actual samples, if available, in the same way as seen classes, or can be simulated in a non-trivial way to represent potentially existing species.

Our approach uses DNA as side information for the first time for zero-shot classification of species. In fine-grained, large-scale species classification, no other side information can explain class dichotomy better than DNA, as new species are explicitly defined based on variations in DNA. The hierarchical Bayesian model leverages the implicit inter-species association of DNA and phenotypic traits and ultimately allows us to establish a Bayesian hierarchy based on DNA similarity between unseen and seen classes. We compare DNA against word representations for assessing class similarity and show that the Bayesian model that uses DNA to identify similar classes achieves favorable results compared to the version that uses word representations on a well-known ZSL benchmark species dataset involving slightly less than 200 bird species. In the particular case of an insect dataset with over 1000 species, when visual attributes or word representations may not offer feasible alternatives, we show that our hierarchical model that relies on DNA to establish class hierarchy significantly outperforms all other embedding-based methods and feature generating networks.

Our contributions are on three fronts. First, we introduce DNA as side information for fine-grained ZSL tasks, implement a Convolutional Neural Net (CNN) model to learn DNA barcode embeddings, and show that the embeddings are robust and highly specific for closed-set classification of species, even when training and test sets of species are mutually exclusive. We use the benchmark CUB dataset as a case study to show that DNA embeddings are competitive to word embeddings as side information. Second, we propose a fine-grained insect dataset involving $21,212$ matching image/DNA pairs from $578$ genera and $1,213$ species as a new benchmark dataset and discuss the limitations of current ZSL paradigms for fine-grained ZSL tasks when there is no strong association between side information and image features. Third, we perform extensive studies to show that a simple hierarchical Bayesian model that uses DNA as side information outperforms state-of-the-art ZSL techniques on the newly introduced insect dataset by a wide margin.

## 2   Related Work

**Zero-Shot Learning.**   Early ZSL literature is dominated by methods that embed image features into a semantic space and perform various forms of nearest neighbor search to do inference (14; 41; 1). As the dimensionality of semantic space is usually much smaller than the feature space this leads to the

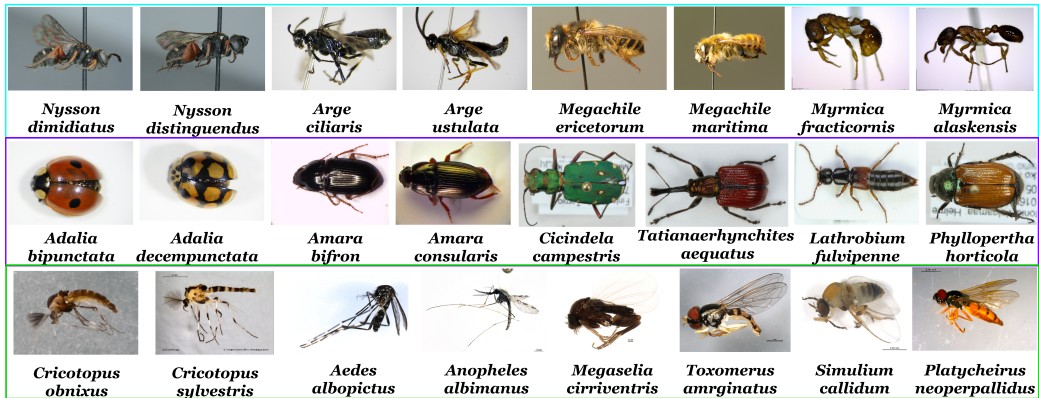

Figure 1: Image samples from the INSECT dataset. Rows represents a small subset of species from three orders: Hymenoptera, Coleoptera and Diptera, respectively. The first word in names indicate genus, the two words together define the species name.

hubness problem where some classes become *hub* and occur as the nearest neighbor of many samples. In an effort to alleviate the hubness problem, (50; 40) changed the direction of the embedding from semantic to image feature space. This was followed by a line of work that investigates bidirectional embedding between semantic and image spaces through a latent space (51; 43; 2; 32; 38).

In (25; 15), a new strategy of synthesizing features for unseen classes and converting the challenging ZSL problem into traditional supervised learning is introduced (23; 44; 9; 13; 48; 52; 29; 39; 4). Although feature generating networks (FGNs) currently achieve state-of-the-art results in ZSL, they suffer from the same problem as earlier lines of work in ZSL: hypersensitivity towards side information not strongly correlated with visual attributes. The vulnerability of both embedding and FGN-based methods toward sources of side information different than visual attributes, such as word vectors or WordNet hierarchy, is investigated in (2; 39; 44). Another limitation of FGNs is that features generated for unseen classes are significantly less dispersed than actual features due to the generator failing to span more than a small subset of modes available in the data. Recent deep generative models mitigate this problem by proposing different loss functions that can better explore inter-sample and inter-class relationships (3; 7; 8; 19; 45). However, these methods fail to scale well with an increasing number of classes with an especially high inter-class similarity (24).

**Side Information in ZSL.** Side information serves as the backbone of ZSL as it bridges the knowledge gap between seen and unseen classes. Earlier lines of work (22; 1) use visual attributes to characterize object classes. Although visual attributes achieve compelling results, obtaining them involves a laborious process that requires manual annotation by human experts not scalable to data sets with a large number of fine-grained object classes. When dealing with fine-grained species classification, apart from scalability, a more pressing obstacle is how to define subtle attributes potentially characteristic of species that have never been observed.

As an alternative to manual annotation, several studies (11; 14; 2; 46; 34; 5) proposed to learn side information that requires less effort and minimal expert labor such as textual descriptions, distributed text representations, like Word2Vec (27) and GloVe (33), learned from large unsupervised text corpora, taxonomical order built from a pre-defined ontology like WordNet (28), or even human gaze reaction to images (20). The accessibility, however, comes at the cost of performance degradation (2; 39). A majority of ZSL methods implicitly assume strong correlation between side information and image features, which is true for handcrafted attributes but less likely to be true for text representations or taxonomic orders. Consequently, all these methods experience significant decline in performance when side information is not based on visual attributes.

## 3   Barcode of Life Data and DNA Embeddings

In this study, we present the fine-grained INSECT dataset with $21,212$ matching image/DNA pairs from $1,213$ species (see Fig. 1 for sample images). Unlike existing benchmark ZSL datasets, this new

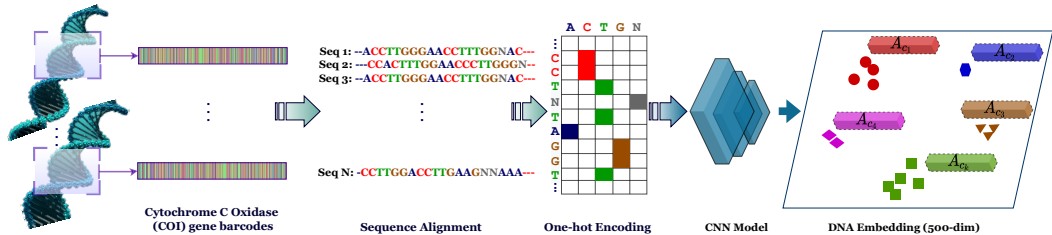

Figure 2: Attribute extraction from mitochondrial DNA.

dataset uses DNA as side information[1] and can be best characterized with the high degree of similarity among classes. Among the existing benchmark datasets, SUN contains the largest number of classes (717) but classes in SUN represent a wide range of scene categories related to transportation, indoor and outdoors, nature, underwater etc., and as such can be considered a relatively coarse-grained dataset compared to the INSECT dataset we are introducing in this study.

All insect images and associated DNA barcodes in our dataset come from the Barcode of Life Data System (BOLD) (35; 36). BOLD is an open-access database in which users can upload DNA sequences and other identifying information for any living organism on Earth. The database provides approximately 658 base pairs of the mitochondrial DNA barcode extracted from the *cytochrome c oxidase I* (COI) gene along with additional information such as country of origin, life-stage, order, family, subfamily, and genus/species names.

**Data Collection.** We collected image/DNA pairs of insects that originate from three orders: Diptera (true flies), Coleoptera (beetles) and Hymenoptera (sawflies, wasps, bees, and ants). While the dataset is in general clean, manual effort was devoted to further curate the dataset. Only cases with images and matching DNA barcodes of adult insects are included. Images from each species were visually inspected and poor quality images were deleted. Only species with more than ten instances were included. The final dataset consisted of $21,212$ images and $1,213$ insects species of which $254$ belong to Diptera (133 genera), $564$ to Coleoptera (315 genera) and $395$ to Hymenoptera (130 genera). We extracted image features, namely image embeddings, using a pre-trained (on ImageNet 1000 classes) ResNet101 model (16). Images are resized to $256 \times 256$ and center-cropped before fed to the ResNet model. No other pre-processing is applied to the images.

**Data Split.** We randomly chose 10% of all species as unseen classes for the test set leading to $1,092$ seen and 121 unseen classes. Similarly, we randomly chose 10% of the $1,092$ training classes as unseen classes for the validation set. Samples from seen classes were split by a $80/20$ ratio in a stratified fashion to create seen portion of the train and test datasets. In the dataset there were a few hundred cases where multiple image views (dorsal, ventral, and lateral) of the same insect were present. To avoid splitting these cases between train and test, we made sure all instances of the same insect are included in the

|  | $Y^{all}$ | $Y^s$ | $Y^u$ |
|---|---|---|---|
| #Images | 21,212 | 3,525 | 2,425 |
| #Classes | 1,213 | 1,080 | 121 |

Table 1: ZSL split details. $Y^s$, and $Y^u$ denote the seen and unseen test sets, whereas $Y^{all}$ represents entire data. There are $15,262$ $(21,212 - 3,525 - 2,425)$ samples left for the training set.

training set. As a result, 12 of the $1,092$ seen classes in the training set were not represented in the test set. Our dataset splits are summarized in Table 1.

**DNA Embeddings.** Although it is the first time DNA barcodes are used as side information in ZSL domain, there have been some work investigating vector embeddings for DNA sequences. Authors of (31) trained a CNN model to do binary DNA sequence classification considering sequences as a text data. Imitating amino acid structure, each triplet of base pairs are treated as a word and sequences are converted into one-hot vector representation. Taking (27) as the base, (30) trained a shallow neural network on human genome data to generate representation for k-mers. Unlike these techniques we deal with DNA Barcodes represented by nucleotide sequences and aim to convert the entire character sequence into a vector embedding useful for species classification with more than 1,000 classes.

---

[1]Please refer to supplementary material for discussion on limitations of using DNA as side information

Most recently, DNABERT (18) adapted the powerful text transformer model (10) to a genomic DNA setting and generated vector embeddings for long DNA sequences.

In this paper, we trained a CNN model to learn a vector representation of DNA barcodes in the Euclidean space. First, the consensus sequence of all DNA barcodes in the training set with 658bp is obtained. Then, all sequences are aligned with respect to this consensus sequence using a progressive alignment technique implemented in MATLAB R2020A (Natick, MA, USA). A total of five tokens are used, one for each of the four bases, *Adenine*, *Guanine*, *Cytosine*, *Thymine*, and one for *others*. All ambiguous and missing symbols are included in the *others* token. In pre-processing, barcodes are one hot encoded into a 658x5 2D array, where 658 is the length of the barcode sequence (median of the nucleotide length of the DNA data).

To train the CNN model, a balanced subset of the training data is subsampled, where each class size is capped at 50 samples. The CNN is trained with $14,723$ barcodes from $1,092$ classes. No barcodes from the 121 unseen classes are employed during model training. The training set is further split into two as train ($80\%$) and validation ($20\%$) by random sampling. We used 3 blocks of convolutional layers each followed by batch normalization and 2D max-pooling. The output of the third convolutional layer is flattened and batch normalized before feeding the data into a fully-connected layer with 500 units. The CNN architecture is completed by a softmax layer. We used the output of the fully-connected layer as the embeddings for DNA. Class level attributes are computed by the mean embedding of each class. The DNA-based attribute extraction is illustrated in Figure 2. The details of the model architecture is depicted in Figure 3 in Supplementary material. We used ADAM optimizer for training the model for five epochs with a batch size of 32 (with a step-decay initial learning rate = 0.0005 and drop factor= 0.5, $\beta_1 = 0.9$, $\beta_2 = 0.999$). The model is developed in Python with Tensorflow-Keras API.

**Predictive accuracy of DNA embeddings.** Although the insect barcodes we used are extracted from a single gene (COI) of the mitochondrial DNA with a relatively short sequence length of 658 base pairs, they are proven to have exceptional predictive accuracy; the CNN model achieves a 99.1% accuracy on the held-out validation set. Note that, we only used the data from training seen classes to train the CNN model. In order to validate the generalizability of embeddings to unseen data, we trained a simple K-Nearest Neighbor classifier ($K = 1$) on the randomly sampled $80\%$ of the DNA-embeddings of unseen classes and tested on the remaining $20\%$. The classifier had a perfect accuracy for all 121 but one classes with an overall accuracy of 99.8%.

In addition to our CNN model we have explored a DNABERT (18) model for converting DNA barcodes to vector embeddings. The pretrained DNABERT model achieves around 85% (vs 99% from CNN) top-1 KNN accuracy (averaged over 10 runs) on the unseen classes. Pretrained DNABERT can be fine-tuned for species classification however because of the vast number of parameters to tune each run takes a few hours on a relatively sophisticated GPU, significantly more than CNN training. Similarly, a simple LSTM model with half of the parameters as the CNN model is almost 5 times slower than the CNN model and requires more epochs to reach a reasonable accuracy. Therefore, we use a simple 3-layer CNN that trains in an hour and achieves almost perfect top-1 KNN accuracy.

To demonstrate that the approach can be easily extended to larger members of the animal kingdom, we compiled approximately $26,000$ DNA barcodes from $1,047$ bird species to train another CNN model (ceteris paribus) to learn the DNA embeddings for CUB dataset (see the Supp. materials for details). The CNN model achieved a compelling 95.60% on the held-out validation set.

## 4 Bayesian Zero-shot Learning

Object classes in nature already tend to emerge at varying levels of abstraction, but the class hierarchy is more evident when classes represent species and species are considered the lowest taxonomic rank of living organisms. We build our approach on a two layer hierarchical Bayesian model that was previously introduced and evaluated on benchmark ZSL datasets with promising results (6). The model assumes that there are latent classes that define the class hierarchy in the image space and uses side information to build the Bayesian hierarchy around these latent classes. Two types of Bayesian priors are utilized in the model: global and local. As the name suggests, global priors are shared across all classes, whereas local priors represent latent classes, and are only shared among similar classes. Class similarity is evaluated based on side information in the Euclidean space. Unlike

standard Bayesian models where the posterior predictive distribution (PPD) forms a compromise between prior and likelihood, this approach utilizes posterior predictive distributions to blend local and global priors with data likelihood for each class. Inference for a test image is performed by evaluating posterior predictive distributions and assigning the sample to the class that maximizes the posterior predictive likelihood.

**Generative Model.**    The two-layer generative model is given below,

$$\boldsymbol{x_{jik}} \sim N(\boldsymbol{\mu_{ji}}, \Sigma_j), \quad \boldsymbol{\mu_{ji}} \sim N(\boldsymbol{\mu_j}, \Sigma_j \kappa_1^{-1}), \quad \boldsymbol{\mu_j} \sim N(\boldsymbol{\mu_0}, \Sigma_j \kappa_0^{-1}), \quad \Sigma_j \sim W^{-1}(\Sigma_0, m) \tag{1}$$

where $j, i, k$ represent indices for local priors, classes, and image instances, respectively. We assume that image feature vectors $\boldsymbol{x_{jik}}$ come from a Gaussian distribution with mean $\boldsymbol{\mu_{ji}}$ and covariance matrix $\Sigma_j$, and are generated independently conditioned not only on the global prior but also on their corresponding local priors.

Each local prior is characterized by the parameters $\boldsymbol{\mu_j}$ and $\Sigma_j$. $\boldsymbol{\mu_0}$ is the mean of the Gaussian prior defined over the mean vectors of local priors, $\kappa_0$ is a scaling constant that adjusts the dispersion of the means of local priors around $\boldsymbol{\mu_0}$. A smaller value for $\kappa_0$ suggests that means of the local priors are expected to be farther apart from each other whereas a larger value suggests they are expected to be closer. On the other hand, $\Sigma_0$ and $m$ dictate the expected shape of the class distributions, as under the inverse Wishart distribution assumption the expected covariance is $E(\Sigma|\Sigma_0, m) = \frac{\Sigma_0}{m-D-1}$, where $D$ is the dimensionality of the image feature space. The minimum feasible value of $m$ is equal to $D + 2$, and the larger the $m$ is the less individual covariance matrices will deviate from the expected shape. The hyperparameter $\kappa_1$ is a scaling constant that adjusts the dispersion of the class means around the centers of their corresponding local priors. A larger $\kappa_1$ leads to smaller variations in class means relative to the mean of their corresponding local prior, suggesting a fine-grained relationship among classes sharing the same local prior. Conversely, a smaller $\kappa_1$ dictates coarse-grained relationships among classes sharing the same local prior. To preserve conjugacy of the model, the proposed model constrains classes sharing the same local prior to share the same covariance matrix $\Sigma_j$. Test examples are classified by evaluating posterior predictive distributions (PPD) of seen and unseen classes. As illustrated in Fig. 3 the PPD in general incorporates three sources of information: the data likelihood that arises from the current class, the local prior that results from other classes sharing the same local prior as the current class, and global prior defined in terms of hyperparameters. PPDs for seen classes include the global prior and data likelihood and are derived in the form a Student-t distribution whereas for unseen classes the data likelihood does not exist as no image samples are available for these classes. We leave the details of derivations to the supplementary material.

**Surrogate classes.**    According to the generative model in (1), groupings among classes are determined based on local priors. Thus, once estimated from seen classes, local priors can be used to define surrogate classes for unseen ones during inference. Associating each unseen class with a unique local prior forms the basis of our approach. The local prior for each unseen class is defined by finding the $K$ seen classes most similar to that unseen class. The similarity is evaluated by computing the $\mathcal{L}^2$ (*Euclidean*) distance between class-level attribute or embedding vectors ($\phi$) obtained from the side information available. Once a local prior is defined for each unseen class the PPD for the corresponding surrogate class can be derived in terms of only global and local priors as in equation (2). Test examples are classified based on class-conditional likelihoods evaluated for both seen and surrogate classes.

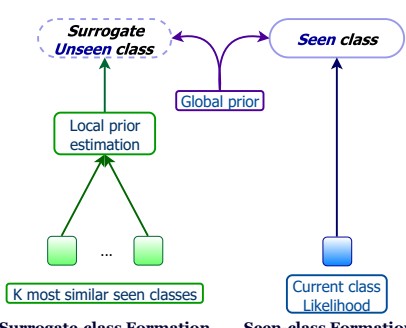

Figure 3: Class formations for PPD during inference.

$$P(\boldsymbol{x}|\{\bar{\boldsymbol{x}}_{\boldsymbol{ji}}, S_{ji}\}_{t_i=j}, \boldsymbol{\mu_0}, \kappa_0, \kappa_1) = T(\boldsymbol{x}|\bar{\boldsymbol{\mu}}_{\boldsymbol{j}}, \bar{\Sigma}_j, \bar{v}_j); \qquad \bar{\boldsymbol{\mu}}_{\boldsymbol{j}} = \frac{\sum_{i:t_i=j} \frac{n_{ji}\kappa_1}{(n_{ji}+\kappa_1)} \bar{\boldsymbol{x}}_{\boldsymbol{ji}} + \kappa_0 \boldsymbol{\mu_0}}{\sum_{i:t_i=j} \frac{n_{ji}\kappa_1}{(n_{ji}+\kappa_1)} + \kappa_0},$$

$$\bar{v}_j = \sum_{i:t_i=j} (n_{ji} - 1) + m - D + 1, \quad \bar{\Sigma}_j = \frac{(\Sigma_0 + \sum_{i:t_i=j} S_{ji})(\tilde{\kappa}_j + 1)}{\tilde{\kappa}_j \bar{v}_j} \tag{2}$$

where, $\bar{\boldsymbol{x}}_{\boldsymbol{ji}}, S_{ji}$ and $n_{ji}$ represent sample mean, scatter matrix and size of class $i$ associated with local prior $j$, respectively and $\tilde{\kappa}_j$ is defined as in Eq. (30) in the supplementary material[2].

**Rationale for the hierarchical Bayesian approach and limitations.** We believe that the hierarchical Bayesian model is ideally suited for fine-grained zero-shot classification of species when DNA is used as side information for the following reasons. The performance of the model in identifying unseen classes depends on how robust the local priors can be estimated. This in turn depends on whether or not the set of seen classes contain any classes similar to unseen ones. As the number of seen classes increases, seen classes become more representative of their local priors, more robust estimates of local priors can be obtained, and thus, unseen classes sharing the same local priors as seen classes can be more accurately identified. On the other hand, if the class-level side information is not specific enough to uniquely characterize a large number of classes, then the model cannot evaluate class similarity accurately and local priors are estimated based on potentially incorrect association between seen and unseen classes. In this case having a large number of seen classes available may not necessarily help. Instead, highly specific DNA as side information comes into play for accurately evaluating class similarity. If a unique local prior can be eventually described for each unseen class, then unseen classes can be classified during test time without the model having to learn the mapping between side information and image features beforehand. Uniqueness of the local prior can only be ensured when the number of seen classes is large compared to the number of unseen classes. Thus, the ratio of the number of seen and unseen classes becomes the ultimate determinant of performance for the hierarchical Bayesian model. The higher this ratio is the higher the accuracy of the model will be. An experiment demonstrating this effect is performed in Section 5.3.

If the same set of $K$ classes is found to be the most similar for two different unseen classes, then these two unseen classes will inherit the same local prior and thus they will not be statistically identifiable during test time. The likelihood of such a tie happening for fine-grained data sets quickly decreases as the number of classes increases. In practice we deal with this problem by replacing the least similar of the $K$ most similar seen classes by the next most similar seen class for one of the unseen classes.

## 5 Experiments

In this section we report results of experiments with two species datasets that use DNA as side information. Details of training and hyperparameter tuning are provided in the supplementary material along with the source code of our methods.

### 5.1 Experiments with the INSECT dataset

We compare our model (BZSL) against state-of-the-art (SotA) ZSL methods proved to be most competitive on benchmark ZSL datasets that use visual attributes or word vector representations as side information. Selected SotA models represent various ZSL categories: (1) Embedding methods with traditional (1; 37) and end-to-end neural network (49) approaches, (2) FGNs using VAE (39) and GAN (44), and (3) end-to-end few shot learning approach extended to ZSL (43). Table 2 displays seen (**S**) and unseen (**US**) accuracies and their harmonic mean (**H**) on the

| Method | US | S | H |
|---|---|---|---|
| CRNet (49) | 13.33 | 19.70 | 15.90 |
| ALE (1) | 2.86 | 27.18 | 5.17 |
| RelationNet (43) | 3.25 | 24.37 | 5.73 |
| CADA-VAE (39) | 14.55 | 20.81 | 17.10 |
| ESZSL (37) | 3.41 | 18.61 | 5.77 |
| LsrGan (44) | 12.58 | 30.41 | 17.75 |
| **BZSL** | **20.83** | **38.30** | **26.99** |

Table 2: Generalized ZSL results on Insect data using DNA barcodes as attributes.

INSECT data using DNA as the side information. Results suggest that the large number of seen classes along with the highly specific nature of DNA information in characterizing classes particularly favors the Bayesian method to more accurately estimate local priors and characterize surrogate classes. The harmonic mean achieved by the proposed method is 52% higher than the harmonic mean achieved by the second best performing technique. Similar levels of improvements are maintained on both seen and unseen class accuracies. The next top performers are FGNs. CADA-VAE uses a VAE whereas LsrGan utilizes GAN to synthesize unseen class features, then both train a *LogSoftmax* classifier for inference. Lower unseen class accuracies suggest that FGNs struggle to synthesize meaningful

---

[2]The code and dataset are available at `https://github.com/sbadirli/Fine-Grained-ZSL-with-DNA`

| Method | Attributes | | | Word Vectors | | | DNA | | |
|---|---|---|---|---|---|---|---|---|---|
| | US | S | H | US | S | H | US | S | H |
| CRNet (49) | 44.28 | 59.84 | 50.89 | 22.75 | 45.92 | 30.43 | 9.27 | 56.56 | 15.93 |
| ALE (1) | 25.15 | 60.80 | 35.59 | 3.95 | 48.57 | 7.31 | 3.50 | 50.18 | 6.54 |
| RelationNet (43) | 11.66 | 44.81 | 18.50 | 8.67 | 36.16 | 13.99 | 5.33 | 40.83 | 9.42 |
| CADA-VAE (39) | 47.15 | 53.11 | 49.95 | 26.45 | 41.98 | **32.45** | 19.42 | 37.05 | 25.48 |
| ESZSL (37) | 15.58 | 50.66 | 23.84 | 2.26 | 23.86 | 4.12 | 5.99 | 5.38 | 5.67 |
| LsrGan (44) | 47.65 | 56.97 | **51.89** | 24.63 | 37.96 | 29.88 | 15.99 | 33.57 | 21.66 |
| **BZSL** | 31.49 | 50.61 | 38.82 | 22.43 | 45.00 | 29.94 | 27.46 | 48.14 | **34.97** |

Table 3: Generalized ZSL results on CUB data using original visual attributes, word vectors, and DNA attributes. **US**, **S**, and **H** represent unseen, seen class accuracies and harmonic mean, respectively.

features in the image space. On the other hand, CRNet that uses end-to-end neural network to learn the embedding between semantic and image spaces renders slightly worse performance than FGNs. It seems, non-linear embedding also works better than a linear (ESZSL) and bilinear (ALE) ones for this specific dataset. RelationNet is amongst the ones with the lowest performance, as the method is explicitly designed for Few-shot learning and expects the side information to be strongly correlated with image features. The weak association between side information and image features affects the performance of both FGNs and embedding methods, but the traditional embedding methods suffer the most.

## 5.2 Experiments with the benchmark CUB dataset

To demonstrate the utility of DNA-based attributes in a broader spectrum of species classification, we procured DNA barcodes, again from the BOLD system, for bird species in the CUB dataset. For this experiment, we derived 400 dimensional embeddings in order to have the same size with word vectors and eliminate the attribute size effect. There were 6 classes, 4 seen and 2 unseen, that did not have DNA barcodes extracted from COI gene in the BOLD system. These classes were excluded from the dataset but the proposed split from (47) is preserved otherwise.

The results shown in Table 3 validate our hypothesis that when side information is not strongly correlated with visual characteristics of object classes (like in word vectors or DNA) both embedding methods and FGNs display significant performance degradation. With the exception of the proposed Bayesian model, word vector representation yields better accuracy than DNA-based attributes for all models. This phenomenon can be explained by our observation that text fragments related to common animals/birds in the Wikipedia/Internet often include some morphological traits of the underlying species. Hence, word vector representation is expected to have higher degree of correlation to visual attributes than DNA information. Our model produces the best results, $34.97\%$ vs $32.45\%$ when the side information is not derived from visual characteristics of classes. This outcome validates the robustness of the Bayesian model to diverse sources of side information and emphasizes the need for more robust FGN or embedding based models in more realistic scenarios where hand-crafted visual attributes are not feasible.

## 5.3 The effect of the number of seen classes on performance

Local priors are central to the performance of the hierarchical Bayesian model. Here, we perform experiments to show that as the number of seen classes increases while the number of unseen classes is fixed, each unseen class can be associated with a larger pool of candidate seen classes and more informative local priors can potentially be obtained, which in turn leads to more accurate identification of unseen classes. To demonstrate this effect we run two experiments. In the first experiment we use the same set of unseen classes as in Section 5.1 but gradually increase the number of seen classes used for training. In the second experiment we double the size of the unseen classes and gradually include the remaining classes into training as seen classes. The first experiment is also performed for CADA-VAE. LsrGan is skipped for this experiment due to long training time. To account for random subsampling of seen classes each experiment is repeated five times and error bars are included in each plot. There is a clear trend in these results that further highlights the intuition behind the hierarchical Bayesian model and explains why this model is well-suited for fine-grained ZSL. When $10\%$ of the

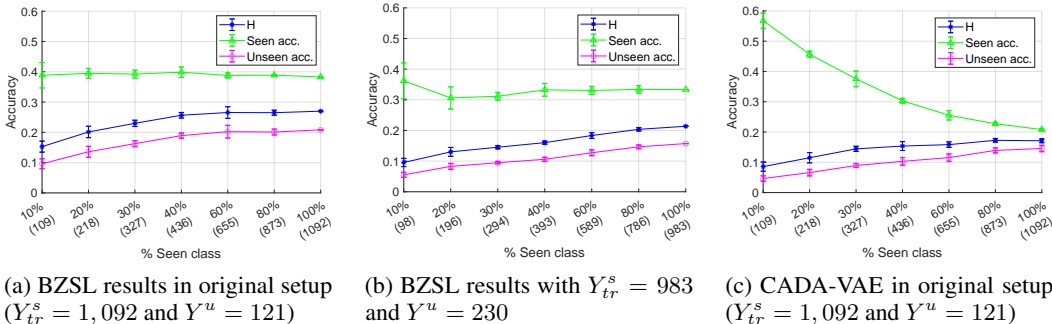

(a) BZSL results in original setup ($Y_{tr}^s = 1,092$ and $Y^u = 121$)

(b) BZSL results with $Y_{tr}^s = 983$ and $Y^u = 230$

(c) CADA-VAE in original setup ($Y_{tr}^s = 1,092$ and $Y^u = 121$)

Figure 4: The effect of the number of seen classes on the performance of BZSL and CADA-VAE. Each experiment is repeated five times to account for random subsampling of seen classes.

classes are used as unseen, unseen class accuracy improves with increasing number of seen classes until it flatlines beyond the $60\%$ mark while seen class accuracy always maintains around the same level (see Fig. 4a). When $20\%$ of the classes are used as unseen no flatlining effect in unseen class accuracy is observed even at $100\%$ mark, which suggest that there is still room for improvement in unseen class accuracy if more seen classes become available (see Fig. 4b). For CADA-VAE unseen class accuracy initially improves and then flatlines beyond $80\%$ mark but this improvement comes at the expense of significant degradation in seen class accuracy, which suggest that as the number of seen classes increase generated features further confound the classifier as would be expected of an FGN for a fine-grained dataset.

## 6   Conclusions

For the first time in the ZSL literature we use DNA as a side information and demonstrate its utility in evaluating class similarity for the purpose of identifying unseen classes in a fine-grained ZSL setting. On the CUB dataset, despite being trained with less than 30,000 very short sequences, we find DNA embeddings to be highly competitive with word vector representations trained on massive text corpora. We emphasize the importance of DNA as side information in zero-shot classification of highly fine-grained species datasets involving thousands of species, and on the INSECT dataset, show that a simple Bayesian model that readily exploits inherent class hierarchy with the help of DNA can significantly outperform highly complex models. We show that SotA ZSL methods that take the presence of an explicit association between visual attributes and image features for granted, suffer significant performance degradation when non-visual attributes such as word vectors and WordNet are used as side information. The same effect is observed with DNA embeddings as well. Although visual attributes tend to be the best alternative as side information for a coarse-grained species classification task, they quickly lose their appeal with an increasing number of classes. Considering the tens of thousands of *described* species and even larger number of *undescribed* species, DNA seems to be the only feasible alternative to side information for large-scale, fine-grained zero-shot classification of species.

These favorable results by a simpler model suggest that as the number of classes increases along with inter-class similarity, the complexity of the mapping between side information and image attributes emerges as a major bottleneck at the forefront of zero-shot classification. A promising future research avenue appears to be implementing hierarchically organized FGNs where each subcomponent only operates with a small subset of seen classes all sharing the same local prior.

This work does not present any foreseeable negative societal consequences beyond those already associated with generic machine learning classification algorithms.

**Acknowledgements:**   This research was sponsored by the National Science Foundation (NSF) under Grant Number IIS-1252648 (CAREER). This work has been partially funded by the ERC (853489 - DEXIM) and by the DFG (2064/1 – Project number 390727645). GM acknowledges support from NSF-ATD grant 2124313. The content is solely the responsibility of the authors and does not necessarily represent the official views of NSF.

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
