# OpenReview forum: "Fine-Grained Zero-Shot Learning with DNA as Side Information"
_NeurIPS.cc/2021/Conference — NeurIPS 2021 Poster_

### Official Review · Reviewer_wkgT · 2021-07-12

**Rating:** 5
**Confidence:** 4

**Summary:**

This paper proposes to use DNA as the side information for zero-shot species classification. The idea is novel and biologically sound. Experiments show that DNA embeddings achieve comparable performance as text embeddings when serving as side information.

**Limitations And Societal Impact:**

I think the proposed idea is novel and interesting. The paper can be greatly improved if the paper is re-organized and more experiments are conducts.

**Main Review:**

Advantages:
1. This paper is the first work that utilizes DNA as side information for Zero-shot species classification, and the authors clearly explain the reasons and insights behind this.
2. This paper introduces a new dataset for Zero-shot species classification and provides codes to reproduce their experiments.

Weaknesses:
1. On the technique side, the paper's main contribution is developing a CNN model to calculate DNA embeddings. However, there are many existing works (e.g., dna2vec, Gene2vec, DNABERT, etc.) that calculate embeddings for DNA sequences. This paper doesn't mention, use or compare with them. Thus, it is unclear how novel/effective the proposed model is.

2. The paper is not well-written and relatively hard to follow.
    -  It is not well-organized. In section 3, the authors introduce data collection, data processing, DNA embedding calculation, and some experimental details and results on DNA embedding, while in Section 4, they introduce the rest part of the proposed method. It may be better to re-organize this paper.
    - The Experimental setting and baseline models are not clearly explained. The tables are hard to understand. It can be very helpful to include more information in the table caption (e.g., explain the experimental setting, evaluation metrics, and meaning of the abbreviations).
    - Other issues:
        - Table 3 should appear after Table 2.
        - Caption of a table should appear above it.
        - The most important reference "[5]" that the proposed model is mainly based on is missing.
        - ......


3. As mentioned in Line 189-190, the Bayesian Zero-shot Learning introduced in Section 4 is previously proposed in the missing reference. It is unclear how this paper relates to that one.


**Time Spent Reviewing:**

3

---

> ### Author Response · Authors · 2021-08-09
> **Our Response to Reviewer wkgT**
>
> > On the technique side, the paper's main contribution is developing a CNN model to calculate DNA embeddings. However, there are many existing works (e.g., dna2vec, Gene2vec, DNABERT, etc.) that calculate embeddings for DNA sequences. This paper doesn't mention, use or compare with them. Thus, it is unclear how novel/effective the proposed model is.
>
> As explained in the last paragraph of the introduction, our main contribution is to introduce DNA as a side information for fine-grained species classification in Zero-Shot Learning. ZSL is a natural task for evaluating the utility of side information. However, DNA has not been explored as side information for this task before.  To be able to utilize DNA as side information, as additional contributions, we have (1) provided the DNA of the existing bird species in the CUB dataset to the community, (2) curated a new dataset, i.e. INSECT, with 21212  images and 1213 species with DNA sequence information. Our code and data will be made publicly available upon acceptance. Other reviewers seem to agree with us on the novel aspects of our work. Reviewer SAgn: "In general, it is interesting to see that DNA data is helpful for ZSL, and a simple Bayesian model performs well on the DNA data."  Reviewer 2AZs: “As of my knowledge this is the first paper proposing to use DNA as side information for zero-shot image classification.”, "An added value is also the INSECT dataset." Reviewer fQEh: "This paper provides an interesting alternative, DNA, as side information for zero-shot learning."
>
> Converting DNA barcodes into vector embeddings was a task we had to undertake to demonstrate the utility of DNA in identifying similar species in the embedding space. CNN was used as a tool to achieve this conversion. Among the methods cited above Gene2vec uses gene co-expression data to generate embeddings to perform gene analysis.  DNA2Vec is trained on human genome data to generate representation for k-mers. Unlike these techniques we deal with DNA Barcodes represented by nucleotide sequences and aim to convert the entire character sequence into a vector embedding useful for species classification. In addition to CNN we have explored a BERT model for converting DNA barcodes to vector embeddings. The pretrained DNABERT model achieves only  85% top-1 KNN accuracy on the unseen classes whereas for the same task our CNN model (trained only on seen classes) achieves 99% accuracy. DNABERT can be configured for the specific task of species classification but this will require substantial training time because in addition to tuning size of k-mers  DNABERT has tens of other parameters to tune and each run takes a few hours on a relatively sophisticated GPU. We are not convinced that this level of sophistication is justified when we consider the fact that a simple 3-layer CNN trains around an hour and achieves almost perfect top-1 KNN accuracy.
>
> > The paper is not well-written and relatively hard to follow. It is not well-organized. In section 3, the authors introduce data collection, data processing, DNA embedding calculation, and some experimental details and results on DNA embedding, while in Section 4, they introduce the rest part of the proposed method. It may be better to re-organize this paper.
>
> Introducing the INSECT data was one of our main contributions. Hence we decided to highlight it earlier in the text. We would appreciate more specific suggestions for reorganization as the other reviewers found the paper well-written, i.e. quoting SAgn: "The paper is well-written, and all the details of training and datasets are included.", quoting 2AZs: "The exposition is overall clear ..." quoting fQEh: " ... this paper provides a clean formulation of the problem, benchmark dataset, and proposes a methodology for how to use DNA information effectively in zero-shot learning"
> > The Experimental setting and baseline models are not clearly explained. The tables are hard to understand. It can be very helpful to include more information in the table caption (e.g., explain the experimental setting, evaluation metrics, and meaning of the abbreviations).
>
> We will revise  the Table captions as suggested.
>
> > Other issues
>
> Thank you for these suggestions. We will address them in the revised version of our manuscript.
>
> > The most important reference "[5]" that the proposed model is mainly based on is missing.
>
> We have cited the article [5] as Author, Title, In ECCV Workshops 2020 in our references to ensure the anonymity. As indicated in our submission our approach is built on the hierarchical Bayesian model in [5]. The model is not one of the contributions of this NeurIPS submission as indicated in the main paper. Our main contributions are using DNA as side information for fine-grained ZSL together with the INSECT dataset that we have introduced and the DNA information of the other datasets we have constructed.

---

> > ### Comment · Reviewer_wkgT · 2021-08-27
> > **Thank you for your detailed response!**
> >
> > Thank you for your clear explanation!
> >
> > After reading your response and discussing it with other reviewers, I still have a few concerns.
> >
> > 1. Using your own model instead of existing modeling to calculate DNA embedding is fine when illustrating the idea of "use DNA as side information". You may leave the use of other models into future works. However, basic literature reviews should be performed and added to the camera-ready version.
> >
> > 2. Using DNA as side information may be limited. First, this is only applicable to living things. Second, most existing datasets do not come with DNA information. In my opinion, one way that possibly solves the second limitation is providing (or pointing out an existing) database of DNA information of common animals/plants.
> >
> > 3. The writing quality should be improved.
> >
> > 4. If calculating DNA embedding is not one of the contributions of your work, the contribution of this paper may not be sufficient.
> >
> > Fortunately, I think points 1-3 are solvable in the camera-ready version. If all of them are solved, I will change my score from "3" to "5". Otherwise, I would stand by my original score of "3".

---

> > > ### Author Response · Authors · 2021-08-28
> > > **Additional Responses to Reviewer wkgT**
> > >
> > > >Using your own model instead of existing modeling to calculate DNA embedding is fine when illustrating the idea of "use DNA as side information". You may leave the use of other models into future works. However, basic literature reviews should be performed and added to the camera-ready version.
> > >
> > > Thank you for this suggestion. We will extend the related work with the following paragraph.
> > > "There has been a number of recent articles discussing the representation of DNA sequence data.  In [1], DNA k-mer subsequences are embedded using  a two-layer network similar to word2vec.  The authors show that embeddings added together is analogous to nucleotide concatenation.  In [2], gene2vec is introduced, where the word2vec framework is applied to co-expressed gene pairs to generate embeddings. In [3] gene2vec sequences are used as input to a CNN for prediction of mammalian N^6-methyladenosine sites from mRNA.  In [4], bidirectional encoder representations from transformers (BERT) are used to identify transcription factor binding sites, splice sites, and identify functional genetic variants."
> > >
> > > [1]. Ng, Patrick. "DNA2vec: Consistent vector representations of variable-length k-mers." arXiv preprint arXiv:1701.06279 (2017).
> > >
> > > [2] Du, Jingcheng, et al. "Gene2vec: distributed representation of genes based on co-expression." BMC genomics 20.1 (2019): 7-15.
> > >
> > > [3]. Zou, Quan, et al. "Gene2vec: gene subsequence embedding for prediction of mammalian N6-methyladenosine sites from mRNA." Rna 25.2 (2019): 205-218.
> > >
> > > [4]. Ji, Yanrong, et al. "DNABERT: pre-trained Bidirectional Encoder Representations from Transformers model for DNA-language in genome." Bioinformatics 37.15 (2021): 2112-2120."
> > >
> > > > Using DNA as side information may be limited. First, this is only applicable to living things. Second, most existing datasets do not come with DNA information. In my opinion, one way that possibly solves the second limitation is providing (or pointing out an existing) database of DNA information of common animals/plants.
> > >
> > > It is true that DNA can be  used as side information only for species classification. However, there are many practical applications for this, such as edible plant classification, new or remaining animal or plant species identification for ecological conservation to mitigate the effects of global warming, etc. which are extremely important in our current times. Therefore, a reliable "living things” classification systems that use computer vision and biological side information have many important application areas. Furthermore, non-living things that become alive in a living host such as viruses are relevant application domains of DNA-based identification systems as well. In fact, we were surprised when we discovered that DNA was not used as an identifying side information for computer vision before. This is one of the breakthroughs of our research.
> > >
> > > We will expand the discussion on DNA as side information and point readers to existing data repositories (BOLD and GenBank) of animals/plants. There are other DNA databases, but they are typically more taxon specific (i.e. Flybase is for Drosophila, Wormbase for C. elegans, etc.). BOLD is unique in providing images, Genbank does not, and only contains nucleic acids.  BOLD and Genbank are therefore useful for gaining information about a large variety/diversity of species.
> > >
> > > From the models perspective, the hierarchical Bayesian model we use can become a simple yet competitive alternative to more sophisticated ZSL techniques as the number of seen classes increases along with inter-class similarity. This suggests that the hierarchical Bayesian model can be potentially applied to ZSL tasks with other type of side information as long as the side information is reliable enough to accurately assess class similarity in the image embedding space and a large number of seen classes are available to associate with unseen classes, a set-up where local priors can serve as good surrogates for unseen classes.
> > >
> > > > The writing quality should be improved.
> > >
> > > We will incorporate corrections suggested by all reviewers in the final version and consider reorganizing certain sections as needed.
> > >
> > > > If calculating DNA embedding is not one of the contributions of your work, the contribution of this paper may not be sufficient.
> > >
> > > Indeed calculating DNA embedding, i.e. conversion of DNA sequences into vector embeddings by a CNN, is one of the contributions of our work. We did not claim it as a contribution, mainly because of the intuitive nature of this operation. However, this conversion was a key first step that takes us to our main contribution; using the vector embeddings to establish the Bayesian hierarchy and then evaluating DNA as a side information in fine-grained ZSL tasks. We believe using DNA to establish the Bayesian hierarchy and then studying different aspects of this model in dealing with fine-grained ZSL tasks  through extensive experiments can be considered as another important contribution of our paper.

---

> > > > ### Comment · Reviewer_wkgT · 2021-08-29
> > > > **I have updated my score**
> > > >
> > > > Thank you for the follow-up clarification. I have changed my score from "3" to "5".

---

### Official Review · Reviewer_fQEh · 2021-07-14

**Rating:** 7
**Confidence:** 4

**Summary:**

The paper proposes using DNA as side information for fine-grained zero-shot learning of insects, as opposed to visual attribute labels. Their proposed approach establishes a hierarchy over the images using DNA information, via a hierarchical Bayesian model over a learned embedding of DNA barcodes. Their method particularly outshines attribute-label-based approaches on an insect identification application, where visual attributes (from public Wikipedia pages, for example) are sparsely labeled or not available. The provide as an additional contribution their paired DNA/image benchmark dataset containing 21,212 examples of 1,213 insect species.

**Limitations And Societal Impact:**

One potential area of social impact would be risks associated with errors in such a system for insect ID, particularly if those errors are systematically biased in some way, in downstream biodiversity monitoring. It would be nice if the authors dug into these potential risks specifically in downstream insect biodiversity and conservation.

**Main Review:**

This paper provides an interesting alternative, DNA, as side information for zero-shot learning.  The main drawback to the generality and scalability of this method is that it requires paired DNA information to be extracted from the specimen in question, which is not necessarily available for any given sighting - in fact it is most likely only available when digitizing and analyzing large natural history collections or within research labs that collect physical specimens (very common in entomology). That said, when that information is available it should be used, and this paper provides a clean formulation of the problem, benchmark dataset, and proposes a methodology for how to use DNA information effectively in zero-shot learning.

Some insect species, for example Rove Beetles, cannot be visually identified by experts unless they are dissected and the genitalia in inspected under a microscope. The error rates in classification from DNA barcoding can be up to 30%, and museum collections are known to contain labeling errors, particularly for challenging-to-identify species. Do the authors have a sense of the accuracy of their dataset subject to these known challenges in accurate insect ID?

It would be very interesting to see a visualization of the taxonomic distribution of the data, and particularly see the resultant distributions across the 3 orders considered of the random train/test/unseen splits (I’m imagining a taxonomic tree with the different splits colored at the end node, or possibly the proportion of categories in each split at each taxonomic level colored in some way).  It would also be good to show the distribution of examples per class in each split, to see the amount of imbalance represented. Further, it would be interesting to look at the distribution of these specimens across digitizing entities contribution to BOLD, to probe whether any potential correlations exist in digitization practice (camera type, background, DNA extraction method, etc) for certain taxa.

Is the DNA embedding network trained with a species classification loss? I assume so but it isn’t explicit in the text.

How much does the similarity of seen classes affect the local prior for surrogate classes (beyond just the ratio of seen to unseen)?  This would be interesting to explore. Perhaps you could probe how results shift based on the number of seen classes within some distance threshold in embedding space? Or build an explicit split based on taxonomy, with ie held out genera instead of species, to probe the robustness of the model to less “familiar” taxonomic branches?

How much is the performance gain over SOTA based on the fact that those methods were not designed with DNA in mind, but instead visual attributes or word vectors?



Text Suggestions & Nits:

In the introduction, your first sentence “Fine-grained species classification is essential in monitoring biodiversity.” is perhaps not well justified for a machine learning audience. It may help to provide a few examples of how it is used - what happens after you identify the species?

In line 30, “fine-grained classification of species” does this refer to classification generally across data type? Or from images specifically? Throughout these first paragraphs clarification about what data type and problem formulation you are referring to would help.

In line 53: “or can be simulated in a non-trivial way 54 to represent potentially existing species.” Can you further elaborate? How would you simulate the DNA metadata for an image of an unseen species in a robust way in order to provide that side information to your model?

Line 82: give a quick high-level description of the hubness problem

Define US, S, H in the caption of Tables 2&3 so that readers do not need to search the text to understand the table, it’s also a bit odd that table 3 appears before table 2 in the document



**Time Spent Reviewing:**

2

---

> ### Author Response · Authors · 2021-08-09
> **Our Response to Reviewer fQEh**
>
> > The main drawback to the generality and scalability of this method is that it requires paired DNA information to be extracted from the specimen in question, which is not necessarily available for any given sighting
>
> It is true that specialized information is required to use DNA as side information. However, this is not much different than using visual attributes as side information, which also requires specialized information and well-trained taxonomists. For example, for annotating insects with visual attributes one has to go through a long list of morphological traits as described in this document [A Pictorial Key to the Order of Adult Insects](https://extension.entm.purdue.edu/401Book/pdf/order_pictorial_key.pdf). This list does not even cover emerging morphological traits in unseen species. DNA is much more precise than visual  attributes in identifying species, mainly because speciation occurs as a result of changes in DNA.
>
> > The error rates in classification from DNA barcoding can be up to 30%, and museum collections are known to contain labeling errors, particularly for challenging-to-identify species. Do the authors have a sense of the accuracy of their dataset subject to these known challenges in accurate insect ID?
>
> There are certainly classes of insects that contain a lot of biodiversity and they cannot be reliably identified, requiring as mentioned genitalia dissections and likely DNA data. And these errors are then translated into incorrectly labeled DNA sequences. As studied in the following articles Genbank and BOLD databases are known to have some errors (references below).  The limitations are the humans doing the identifications. We believe computer vision has the potential to exceed human identification in its ability to extract subtle characteristics humans cannot ‘see’, especially when better (3D) images are used.
>
> Pentinsaari, Mikko, et al. "BOLD and GenBank revisited–Do identification errors arise in the lab or in the sequence libraries?." PLoS One 15.4 (2020): e0231814.
>
> Meiklejohn, Kelly A., Natalie Damaso, and James M. Robertson. "Assessment of BOLD and GenBank–Their accuracy and reliability for the identification of biological materials." PloS one 14.6 (2019): e0217084.
>
> >It would be very interesting to see a visualization of the taxonomic distribution of the data, and particularly see the resultant distributions across the 3 orders ....
>
> Thank you for this suggestion. As OpenReview does not support illustrations we have prepared a table instead that shows this taxonomical relation and the performance of BZSL model across different orders. Figure 2 in the Supp. material contains histograms of the number of species per genus. Regrettably, the information about digitizing entities is limited to country and institution names. No information is available in BOLD about camera type and extraction methods.
>
> Table: Number of seen and unseen classes in the INSECT dataset at the order level and their corresponding seen and unseen class accuracies.
>
> |		|SEEN		|	|UNSEEN	|	|
> |-------------------|-------------------|---------|-------------------|---------|
> |Order (#training samples) |#test samples | Accuracy | # test samples | Accuracy|
> |Coleoptera (6105)	|1433|49%|450|16%|
> |Diptera (3628)	|874|42%|698|21%|
> |Hymenoptera (4979)	|1198|27%|421|28%|
>
>
>
>
> > Is the DNA embedding network trained with a species classification loss? I assume so but it isn’t explicit in the text.
>
> Thanks for pointing this out. Yes, it is trained by the cross entropy loss. We’ll make this clear in the text.
>
> > How much does the similarity of seen classes affect the local prior for surrogate classes (beyond just the ratio of seen to unseen)?
>
> Thank you for this insightful comment. The train/test splits for the INSECT data introduced in our study is compiled to allow for some of the unseen species to be not represented at the genus level in the training set. This constitutes a relatively challenging scenario because local priors for unseen species not represented at the genus level in the training have to be estimated by seen species of other genera. These will be species that are less similar to each other and the unseen species in the image space. In the INSECT dataset there are 40 unseen species, which are not represented at the genus level in the training set (no seen species available from the same genus in training) and the average accuracy for these unseen classes is only 9%. For unseen species whose genus is represented by one seen species in the training the accuracy jumps to 17%, for those whose genus is represented by two seen species the accuracy is 26%, for those whose genus is represented by three or more seen species the accuracy is 31%. This trend suggests that the performance of BZSL for an unseen class improves as the number of seen classes similar to that unseen class increases during training.
>
> > How much is the performance gain over SOTA based on the fact that those methods were not designed with DNA in mind, but instead visual attributes or word vectors?
>
> Indeed changing the side information requires re-tuning the parameters of all models and we have made sure that we have reevaluated all the methods fairly in this sense. We believe the main limitation that makes SOTA models less competitive is the lack of explicit correlation between mitochondrial DNA and phenotypic traits in species. Unlike visual attributes where direct correlation between individual attributes and phenotypic traits can be established for most attributes, DNA information cannot be explicitly split into different compartments each describing a different phenotypic trait. That is why for fine-grained species classification we expect the improvements gained by SOTA models due to different design choices to be limited when DNA is used as a side information.
>
> > In the introduction, your first sentence “Fine-grained species classification is essential in monitoring biodiversity.” is perhaps not well justified for a machine learning audience. It may help to provide a few examples of how it is used - what happens after you identify the species?
>
> A brief justification will be added to the text.  The biological world is so intertwined with species contributing to ecosystem services, an incomplete understanding of biodiversity has the potential to lead to impacts on an environment that could lead to the extinction of many other species.  In fact, insect species are disappearing faster than they can be identified using traditional means. The identification of new species is reliant on a two step process: first the discovery of a potentially new species, followed by the identification (via the description of attributes that makes it different from other known species).  Afterwards, the biological and ecosystem function of that species can be studied. When environmental changes occur, whether naturally or through human interventions, there can be lasting effects on biodiversity that may not be seen until it is too late.  Imagine an ecosystem that has experienced large shifts in temperatures, so great that a particular species cannot survive. It’s loss then will have an impact on any species that rely on some function in the ecosystem of that species, and they may as a result perish. We are unlikely to see these impacts until many species are lost, and it may not be possible to mitigate these impacts. The sooner biodiversity can be measured, the sooner interventions can be put in place.
>
> > In line 30, “fine-grained classification of species” does this refer to classification generally across data type? Or from images specifically? Throughout these first paragraphs clarification about what data type and problem formulation you are referring to would help.
>
> Thanks for this clarification. That statement refers to images specifically. We will clarify all referrals to data in the revised version.
>
> > How would you simulate the DNA metadata for an image of an unseen species in a robust way in order to provide that side information to your model?
>
> DNA of potential unseen classes can be simulated using a large amount of DNA data and drawing inferences by deep machine learning and traditional bioinformatics methods as to how/where DNA sequences change as speciation occurs, distinguishing between changes leading to speciation and those characterizing individual traits, as well as understanding the degree of these changes at each level of the taxonomy. To model these changes in a robust manner DNA information from a large number of species (>10000) might be needed. With over 200K species and 7M barcodes available in the BOLD repository class insecta can offer an ideal testbed for such a feasibility study.
>
> > Line 82: give a quick high-level description of the hubness problem
>
> A brief description of the hubness problem will be added to the text. A hub class is a class that is more frequently predicted by the classifier for test samples. This usually occurs due to variance shrinkage caused by projecting a higher dimensional image embedding onto a lower dimensional semantic space.
>
> > Define US, S, H in the caption of Tables 2&3 so that readers do not need to search the text to understand the table, it’s also a bit odd that table 3 appears before table 2 in the document
>
> Thanks for these suggestions. We’ll address them in the revised version.
>
> >... It would be nice if the authors dug into these potential risks specifically in downstream insect biodiversity and conservation.
>
> Our method for the initial identification of any insect species has the potential to speed up any biodiversity monitoring, and any results are likely to be verified by taxonomists if conservation efforts are required. Future work in this area is ongoing through validation of the method using samples of a group of taxa (Carabidae) that will be imaged, sequenced, and verified to identify any systematic biases that may exist.

---

> > ### Comment · Reviewer_fQEh · 2021-08-16
> > **Thank you for your detailed response!**
> >
> > I appreciate the clarifications.

---

> ### Comment · Reviewer_fQEh · 2021-08-24
> **Final Recommendation**
>
> After reading the other reviews and the author's feedback, I maintain my original score of 7. That said, I would expect the authors to address the criticisms in the reviews for a camera ready, particularly regarding adding a thorough literature review of other DNA embedding methods to the related work and clarifying and copy-editing the text.

---

### Official Review · Reviewer_2AZs · 2021-07-15

**Rating:** 6
**Confidence:** 3

**Summary:**

Authors propose to use DNA as side information for zero-shot image classification and a method that can leverage this information. The method is based on a Bayessian model that captures similarity between different classes as well as between images of a single class through global and local priors. Unseen classes are represented using the statistics of a fixed number of seen classes that are the most similar according to the side-information. Authors introduce an INSECT database and compare their method on INSECT and CUB with several other zero-shot methods. When using DNA as side information, the proposed method performs the best.

**Limitations And Societal Impact:**

- BSZL performs better than the best SOTA only on the DNA.
- The obvious issue of using DNA as side information is that it is not applicable to non-living objects.

**Main Review:**

As of my knowledge this is the first paper proposing to use DNA as side information for zero-shot image classification. The hierarchical Bayessian approach to model the data is also relevant and I can imagine it becoming one of usual baselines (though it is a pity it is coded in matlab). The exposition is overall clear, find few suggestions below. An added value is also the INSECT dataset.

Suggestions:
- The main text falls short to explain how to arrive from the seen class statistics to computing PPD for an unseen class.
- Table 3 - Include what US/S/H means in the caption.
- Fig. 4 - Use a larger font. Put equal y-axis scale in all three plots.
- Line 76 - "state of the art" => "state-of-the-art"
- Line 133 - "larger" => "more"
- Line 209 - "farther" => "further"
- Line 325 - "fixed" => "stays fixed"
- Line 336 - "maintained" => "maintains"

Final recommendation:
I have read the other reviews and authors responses. I would like thank the authors for the responses and urge them to reflect the criticism in the final paper. I keep my recommendation to accept the paper (with the current score).


**Time Spent Reviewing:**

3

---

> ### Author Response · Authors · 2021-08-09
> **Our Response to Reviewer 2AZs**
>
> > The main text falls short to explain how to arrive from the seen class statistics to computing PPD for an unseen class.
>
> Associating each unseen class with a unique local prior forms the basis of our approach. The local prior for an unseen class is defined by finding the K seen classes most similar to that unseen class. The similarity is evaluated by computing the L2 Euclidean distance between class-level attribute vectors obtained from the side information available such as hand-crafted visual attributes, word vectors or DNA embeddings (as in the case of INSECT data). Once the most similar seen classes are identified the unknown parameters (mean and covariance) of the unseen class are integrated out according to the BZSL generative model to combine local and global priors together in a posterior-predictive distributions. These steps are included in the supplementary material due to space limitations.
>
> > Typos
>
> Thank you for these corrections. We will address them in the revised version of the manuscript and make sure to release a Python version of the hierarchical Bayesian approach alongside the Matlab implementation.
>
> > BSZL performs better than the best SOTA only on the DNA.
>
> It is true that SOTA, mainly represented by feature generating networks (FGNs), may perform better than BZSL when visual attributes are used as side information. However, the success of FGNs mostly depends on the performance of the generator in these networks, and the generator often fails to span more than a small subset of modes available in the data. Recent methods mitigate this problem by proposing different loss functions that can better explore inter-sample and inter-class relationships. However, these methods do not scale well for a large number of classes with high inter-class similarity, a set-up that characterizes fine-grained ZSL. We observe this in Figure 4.c of the paper where CADA-VAE achieves only a marginal improvement in unseen class accuracy at the expense of significant degradation in seen class accuracy as the number of seen classes increases (seen accuracy drops from ~60% to ~20% while unseen accuracy rises from ~5% to ~15% when the number of seen classes increases from 109 to 1092). In general the hierarchical Bayesian model could be a simple yet robust alternative to SOTA in fine-grained ZSL tasks with a large number of seen classes irrespective of the side information used as long as side information can reliably assess class similarity in the image space.

---

### Official Review · Reviewer_SAgn · 2021-07-27

**Rating:** 7
**Confidence:** 3

**Summary:**

This paper proposed to use DNA as side information for the zero-shot learning tasks.
A CNN is used to generate DNA embeddings, and a Bayesian model is used for zero-shot learning.
Experiments are performed on a newly collected fine-grained insect dataset with image and DNA pairs, as well as on the CUB dataset.
Results showed that the Bayesian model outperforms other methods on the insect dataset using DNA as attributes.
On the CUB dataset, using DNA is comparable with using word vectors, but pre-defined attributes still work better.


**Limitations And Societal Impact:**

Some limitations have been discussed in the paper, and there is no negative societal impact.

**Main Review:**

This paper proposed to use DNA for zero-shot learning, unlike prior works using attributes or text descriptions for side information. Results on the collected insect dataset show improvements over other methods. On the CUB dataset, using Bayesian modeling is better for modeling DNA information, but not on visual attributes or word vectors.

In general, it is interesting to see that DNA data is helpful for ZSL, and a simple Bayesian model performs well on the DNA data.
The paper is well-written, and all the details of training and datasets are included. Ablation studies on the number of seen classes and hyper-parameters are also presented.

Questions:
* Why use CNN (on top of one-hot encoding) instead of RNN or other methods for encoding the DNA data?
For the CUB dataset, the DNA embedding is trained on an external dataset of 1047 bird species. Is it a fair comparison between using attributes and word vectors (in Table 3)?
* As stated in L. 246-264, the performance of the Bayesian model is highly correlated to the ratio of the number of seen and unseen classes. If there are similar classes in the seen classes, then the model can perform well. Is this limitation the reason why using visual attributes performs better than using DNA on the CUB dataset?
* Another limitation of the proposed method is the trade-off between seen and unseen classes with different hyper-parameters (Fig. 5) Is there a way to mitigate this, i.e., having good performance on both seen and unseen classes?

Typo:
- Reference [5] is missing
- L.10 "of" CUB dataset

**Time Spent Reviewing:**

4

---

> ### Author Response · Authors · 2021-08-09
> **Our Responses to Reviewer SAgn**
>
> >  Why use CNN (on top of one-hot encoding) instead of RNN or other methods for encoding the DNA data?
>
> Converting DNA barcodes into vector embeddings was a task we had to undertake to demonstrate the utility of DNA in identifying similar species in the embedding space. CNN was used as a tool to achieve this conversion. In addition to CNN we have explored a DNABERT model for converting DNA barcodes to vector embeddings. The pretrained DNABERT model achieves around 85% top-1 KNN accuracy (averaged over 10 different train/test splits with unseen classes) on the unseen classes whereas for the same task CNN model achieves 99% accuracy. Pretrained DNABERT can be fine-tuned for the specific task of species classification but this will require substantial training time because in addition to tuning size of k-mers  DNABERT has tens of other parameters to tune and each run takes a few hours on a relatively sophisticated GPU. In terms of training time a simple LSTM model with half of the parameters as the CNN model is almost 5 times slower than the CNN model and requires more epochs to reach a reasonable accuracy. We are not convinced that this level of sophistication is justified when we consider the fact that a simple 3-layer CNN trains around an hour and achieves almost perfect top-1 KNN accuracy.
>
> > For the CUB dataset, the DNA embedding is trained on an external dataset of 1047 bird species. Is it a fair comparison between using attributes and word vectors (in Table 3)?
>
> Compared to the external datasets containing billions of tokens on which word vectors are trained, the external dataset of 1047 bird species can be considered negligible in size. Despite the small data size we show that robust DNA embeddings that accurately characterize unseen bird species can be obtained. Furthermore, a study in the literature (Akata et al., Evaluation of Output Embeddings for Fine-Grained Image Classification, CVPR 2015) uses specialized bird only text corpus (45 MB of text) to train word vectors but does not achieve much better results than plain Wikipedia.
>
> > As stated in L. 246-264, the performance of the Bayesian model is highly correlated to the ratio of the number of seen and unseen classes. If there are similar classes in the seen classes, then the model can perform well. Is this limitation the reason why using visual attributes performs better than using DNA on the CUB dataset?
>
> For the CUB dataset it is possible that species cluster differently in image and DNA spaces. In other words, class similarity assessed in the DNA space may not align well with the class similarity in the image embedding space. We believe this is neither a limitation of the DNA as a  side information nor the Bayesian approach as a classifier but most likely a limitation of the Resnet model pretrained with an arbitrary set of classes from ImageNet. Color appears as a dominant feature in the visual classification of bird species because different bird species  emerge with different colors of the beak, head, eyes, body, tail, neck, crown, tarsus etc. We suspect that the pretrained ResNet model does not capture other important shape and size-based morphological traits characteristics of bird species as well as the color-based features. As a result, image embeddings from two different bird species sharing the same genus may project farther apart from each other in the image embedding space, especially when they have different color-based traits. For example tsne plot of the CUB Resnet embeddings clusters Scott and Hooded orioles together (yellow/black), whereas Orchard oriole (brown/black), which is from the same genus as the other two species is clustered farther apart from them but closer to Eastern Towhee, which is from a different genus but have similar color patterns (black/white/brown) as Orchard oriole. Visual attributes and image embeddings can still align well in this case because most of the visual attributes used in the CUB dataset are indeed color-based attributes. This problem can be mitigated in future work by using Resnet as a backend to a SimCLR*  model and by training the system end-to-end with only bird images. SimCLR trained with only bird images can better capture non-color morphological traits and at the same time allow image embeddings to be less biased towards seen classes to produce more differentiable image embeddings not only for seen classes but unseen ones as well.
> * Chen, Ting, Simon Kornblith, Kevin Swersky, Mohammad Norouzi, and Geoffrey E. Hinton. "Big Self-Supervised Models are Strong Semi-Supervised Learners." Advances in Neural Information Processing Systems 33 (2020): 22243-22255.
>
> > Another limitation of the proposed method is the trade-off between seen and unseen classes with different hyper-parameters (Fig. 5) Is there a way to mitigate this, i.e., having good performance on both seen and unseen classes?
>
> The effect of the hyperparameters on the seen and unseen class accuracies depends on the underlying structure of the data. For example when $\frac{\kappa_0}{\kappa_1}<<1$ and $\kappa_1<1$ while $\Sigma_0=I$, $\mu_0=0$ we can generate a dataset  where local prior means are well-separated with respect to $\mu_0$ and actual class means are well-separated with respect to local prior means but since the deviation of actual class means with respect to local prior means is not as large as the deviation of local prior means with respect to $\mu_0$ both seen and unseen classes will be statistically identifiable and well separated in the feature space (because of $\frac{\kappa_0}{\kappa_1}<<1$). On top of this if $m>>(d+2)$ that would indicate  all class covariances are similar and the Bayesian classifier is expected to generate near perfect accuracy for both seen and unseen classes because the common covariance (the estimate we use for $\Sigma_0$) will be very similar to individual class covariances. If $m$ is close to $d+2$ while other hyperparameter configurations remain the same such a setting would generate classes with very different class covariances. In that case the performance of the Bayesian model on seen classes will depend on the number of training samples available from each seen class and its performance on unseen classes will depend on the number of seen classes sharing the same local prior as the unseen class as well as the number of samples from those classes. The two layer Bayesian model provides us extreme flexibility to model datasets with different structures but ultimately its performance still depends on first, how well the classes are separated in the feature space, second, how representative the training samples are of the seen class distributions, and third, how representative seen classes are of the local priors.
>
> > Typo (Reference [5] is missing, L.10 "of" CUB dataset)
>
> Thanks for these corrections. We’ll revise the manuscript accordingly.

---

> > ### Comment · Reviewer_SAgn · 2021-09-04
> > **Final Rating**
> >
> > Thanks for the response and clarification. I would like to keep my original score, but I would suggest authors include the results of using different embeddings in the final version and the discussions about the tradeoff between seen and unseen classes.

---

### Author Response · Authors · 2021-08-09
**Summary of the positive comments**

We thank all reviewers for their time reviewing our paper and for the useful feedback provided.

- All reviewers found the Bayesian methodology highly relevant for ZSL.
- Reviewers SAgn, 2AZs, fQEh found the use of DNA as a side information for ZSL novel and interesting.
- Reviewers SAgn, 2AZs, fQEh think the paper is overall well-written.
- Reviewers 2AZs and fQEh found the newly introduced INSECT dataset a valuable contribution.

---

### Author Response · Authors · 2021-08-09
**Contributions**

In our submission, we introduce DNA as a side information to build the Bayesian hierarchy for the Zero-Shot Learning task. In addition, we provide the DNA sequence information for an existing benchmark ZSL dataset as well as curate a new fine-grained INSECT dataset. Our results show that using DNA in this way significantly outperforms the SOTA when a large number of  seen classes (with high inter-class similarity) are available along with side information that can accurately assess species similarity.

---

### Decision · Program_Chairs · 2021-09-27

**Decision:**

Accept (Poster)

**Comment:**

This paper addresses the problem of zero shot learning (ZSL) in the context of fine-grained visual categorization using images from the natural world with (for the first time) DNA as side information. Existing work in this space typically exploits alternative forms of side information such as visual attributes or textual information. Experiments are performed using an existing Bayesian ZSL method on the CUB dataset (with additional DNA information) and a newly proposed dataset of insect images, and show that the models using DNA are competitive and often better than those using only textual side information.

Issues that should be addressed in the final text.
* The current related work text is missing a discussion of alternative methods for learning DNA embeddings. This needs to be remedied. (wkgT)
* Include the already performed comparison to DNABert. This is important as the CNN encoder is listed as one of the contributions of the paper but no comparisons are made to alternative methods for encoding this information.
* Discuss the limitations of DNA side-information e.g. that it is not applicable to all visual categories (wkgT).
* Fix the issues with the writing and paper structure as noted by wkgT.
* Fix the issues with the text as noted by 2AZs and fQEh. e.g. Clarify in the caption for Table 3, what US, S, and H represent.

The reviewers raised legitimate concerns regarding the level of technical contribution, the writing quality, and lack of discussion of other approaches for learning DNA embeddings. The final two of these concerns can be addressed in a revision of the text. Despite these concerns, the reviewers were supportive of the paper, with the most critical raising their score after the discussion. The novelty of the application and the merit of the new dataset were quoted as the reasons for recommending acceptance. The authors are strongly encouraged to address the above issues in the final camera ready text.

As a final note, the authors should be aware that how they cited [5] was not consistent with the NeurIPS guidelines. As the paper was already published it should have been cited with the authors names and treated no differently than any other paper:
"If you need to cite one of your own papers, you should do so with adequate anonymization to preserve double-blind reviewing.  For instance, write “In the previous work of Smith et al. [1]…” rather than “In our previous work [1]...”). If you need to cite one of your own papers that is in submission to NeurIPS and not available as a non-anonymous preprint, then include a copy of the cited submission in the supplementary material and write “Anonymous et al. [1] concurrently show...”)."
https://nips.cc/Conferences/2021/CallForPapers